# INCORPORATING CONTINUOUS DEPENDENCE IMPLIES BETTER GENERALIZATION ABILITY

## ABSTRACT

When applying deep-learning-based solvers to differential equations, a key challenge is how to improve their generalization ability, so that the pre-trained models could be easily adapted to new scenarios of interest. In this paper, inspired by the well-known mathematical statements on the continuous dependence of solutions to ordinary differential equations on initial values and parameters, we make a non-trivial extension of the physics-informed neural networks by incorporating additional information on the continuous dependence of solutions (abbreviated as cd-PINN). Our cd-PINN integrates the advantages of neural operators and Meta-PINN, requiring only a few labeled data while enabling solving ordinary differential equations with respect to new initial values and parameters in a fast and accurate way without fine-tuning. As demonstrated through novel examples like the Logistic model, the Lotka-Volterra model, damped harmonic oscillators and a multiscale model for p53 activation, the accuracy of cd-PINN under those untrained conditions is usually 1-3 orders of magnitude higher than PINN. Meanwhile, the GPU time cost for training in the two approaches is comparable. Therefore, we expect that our cd-PINN would be particularly useful in improving the efficiency and accuracy of deep learning-based solvers for differential equations.

## 1 INTRODUCTION

In recent years, applying various deep-learning algorithms for solving differential equations has attracted increasing attention. Deep-learning-based differential equation solvers are generally considered to have a significant potential to improve computational efficiency (Esmaeilzadeh et al., 2020; Kochkov et al., 2021). One particular notable method is Physics-Informed Neural Networks (PINN)(Raissi et al., 2019), which integrates the governing equations into the loss function to train the model and approximates the solutions to differential equations without discrediting the solution domain. Similarly, the Deep Galerkin Method (DGM)(Sirignano & Spiliopoulos, 2018) uses neural networks to approximate solutions to differential equations, but its loss function is based primarily on the Galerkin residual. The Deep Ritz method(E & Yu, 2018) reformulates the differential equation in a variational form and solves it by minimizing the associated energy. Based on the weak solution form of the differential equations, weak adversarial networks(Zang et al., 2020) parameterize both the weak solution and the test function into the primary neural network and the adversarial neural network, respectively. Additionally, several enhanced methods based on PINN have emerged. Some of these methods focus on decomposing the solution domain, allowing the model to be trained in parallel across multiple GPUs, such as conservative Physics-Informed Neural Networks(cPINN), extended Physics-Informed Neural Networks(xPINN), etc.(Jagtap et al., 2020; Jagtap & Karniadakis, 2020). In particular, gPINN(Yu et al., 2022) incorporate the gradient of the differential equations into the loss function to minimize reliance on residual points and P$^2$INN(Cho et al. (2024)) uses the parameters of the equation as additional encode input so that the model can better solve the CDR equations. Specifically for ODEs, Neural ODEs (Chen et al. (2018); Hu et al. (2022)) learn a continuous dynamic model of the data generation process by embedding neural networks into the ODE systems.

However, these methods treat differential equations with varying parameters and initial values as distinct tasks. When the parameters or initial values change, the model must be re-trained. This will lead to unaffordable computational cost in the face of a large amount of tasks for solving differential equations with diverse parameters and initial values. To address the above issue and enhance the universality of solving different differential equations with deep learning, researchers have begun exploring operator learning, which involves using neural networks to learn the mapping between two infinite-dimensional function spaces.

PDE-Net(Long et al., 2018; 2019) is one of the earliest neural operators, inspired by the finite difference method. It designs a specialized convolution kernel to solve both the forward and inverse problems of differential equations. DeepONet(Lu et al., 2019; Wang et al., 2021), grounded in the universal approximation theorem for operators, learns the mapping of functions to functions, specifically mapping the initial values of PDEs to their solutions. PINO (Li et al., 2024) is the first hybrid approach incorporating data and PDE constraints at different resolutions to learn the solution operator of a given family of parametric PDEs. The Fourier Neural Operator(FNO) (Li et al., 2020a) leverages the fast Fourier transform to perform convolution operations in the Fourier space, enabling it to map input functions to target functions with exceptional performance in high-dimensional and complex systems. The Graph Neural Operator(GNO)(Li et al., 2020b) integrates graph neural networks with operator learning, using graph structures to represent spatial points and their connections, efficiently handling input function mappings on irregular networks.

A notable advantage of these models is that once trained, the prediction time for new applications is nearly negligible. However, training these models usually requires a substantial amount of labeled data, whose quantity and quality determine the model's performance to a large extent. This shortcoming has prompted researchers to explore the integration of meta-learning with PINN algorithms. Meta-learning-based PINN can be categorized into two frameworks: feedforward meta-learning and agnostic meta-learning(MAML). In feedforward-based meta-PINN, the meta-learning model primarily learns how to map the configurations of differential equations to the weight parameters of the PINN model, as seen in approaches like Hyper-PINN and Meta-MgNet(de Avila Belbute-Peres et al., 2021; Chen et al., 2022). On the other hand, the MAML-based meta-PINN aims to learn an efficient initialization for the PINN weight parameters that exhibit strong generalization capabilities. This allows the model to be fine-tuned for a new configuration with only a few rounds of gradient updates. For example, by using the reptile-based method, Liu et al.(Liu et al., 2022) directly learns the initialization of the PINN model, while the MAD-PINN (Huang et al., 2022) implicitly encodes the configurations as additional input to the PINN model, then fine-tuning them to reach a best output. Despite significant advances, these meta-learning methods still face many limitations. They often require longer training time, and the fine-tuning procedure may be computationally expensive when many new configurations are involved.

In this paper, we propose a new method that integrates the advantages of neural operators and Meta-PINN, requiring only a small amount of labeled data while enabling accurate predictions on new configurations without the need for fine-tuning. Unlike Meta-PINN, we treat the solution of parametric differential equations as a single task, rather than separated tasks with different configurations. By incorporating the parameters and initial values as additional input and adding the constraints of continuous dependence of solutions on parameters and initial values into the loss function, we make a non-trivial generalization of PINN (cd-PINN). Our cd-PINN exhibits an outstanding performance on a number of ODE solving tasks involving different combinations of parameters and initial values, whose accuracy under those untrained conditions is usually improved by 1-3 orders of magnitude compared to the vanilla PINN.

## 2 Proposed Methods

### 2.1 Mathematical Foundation

Consider general ordinary differential equations in the following form

$$\frac{d\boldsymbol{u}}{dt} = f(t, \boldsymbol{u}, \boldsymbol{\mu}),$$
$$\boldsymbol{u}(t = t_0) = \boldsymbol{u}_0, \tag{1}$$

where $\boldsymbol{f}(t, \boldsymbol{u}, \boldsymbol{\mu})$ is a parameterized (by $\boldsymbol{\mu} \in V \subset \mathbb{R}^m$) continuous function in $I \times W$, with $t \in I, \boldsymbol{u} \in W$. Here $I$ is an open interval of $\mathbb{R}^1$, and $W$ is a domain of $\mathbb{R}^n$. $\boldsymbol{u}_0$ denotes the value of $\boldsymbol{u}(t)$ at time $t_0$. For example, given the growth model $du/dt = ru^k$, where $r, k \in \mathbb{R}$, we have $\boldsymbol{\mu} = (r, k)$.

With the change of initial values or parameters, the solution of ODEs varies accordingly, which therefore can be represented as $\boldsymbol{u} = \boldsymbol{u}(t, \boldsymbol{u}_0, \boldsymbol{\mu})$. Further suppose the right-hand-side term $f$ is locally Lipschitz continuous with respect to $\boldsymbol{u}$ in $W$, i.e. $|f(t, \boldsymbol{u}_1, \boldsymbol{\mu}) - f(t, \boldsymbol{u}_2, \boldsymbol{\mu})| \leq L\|\boldsymbol{u}_1 - \boldsymbol{u}_2\|$ for $\forall \boldsymbol{u}_1, \boldsymbol{u}_2 \in W$; and for all $t \in I$, $f$ not only is locally Lipschitz continuous but also has a uniform Lipschitz constant $L$. Under these assumptions, we can guarantee the local existence and uniqueness of solutions to the ODEs.

**Theorem 1** (local existence and uniqueness). *For any $t_0 \in I$, $\boldsymbol{u}_0 \in W$, and fixed parameters $\boldsymbol{\mu}$, there exists a positive constant $\alpha > 0$, such that equation 1 admits a unique solution $\boldsymbol{u} = \boldsymbol{u}(t, \boldsymbol{u}_0, \boldsymbol{\mu})$ defined on the interval $t \in [t_0 - \alpha, t_0 + \alpha]$. Furthermore, this solution is continuous with respect to time $t$, and satisfies the initial condition $\boldsymbol{u}(t_0) = \boldsymbol{u}_0$ too.*

During the proof of the local existence and uniqueness of the ODE solution, initial values and parameters both have been fixed. However, if $\boldsymbol{u}_0$ and $\boldsymbol{\mu}$ are allowed to be varied, we arrive at the following well-known results on the continuous dependence of the ODE solution on both initial values and parameters.

Firstly, by fixing parameters $\boldsymbol{\mu}$, we have:

**Theorem 2** (continuous dependence on initial values). *Suppose the solution $\boldsymbol{u}(t, \boldsymbol{x}_0, \boldsymbol{\mu})$ to equation 1 is defined on the interval $[t_0, t_1]$. Then there exists a neighborhood $W_1 \subset W$ around $\boldsymbol{x}_0$, such that for any $\boldsymbol{y}_0 \in W_1$, equation 1 has a unique solution $\boldsymbol{u}(t, \boldsymbol{y}_0, \boldsymbol{\mu})$ that is also defined on the interval $[t_0, t_1]$. Furthermore, for $\forall t \in [t_0, t_1]$, the following inequality holds*

$$\|\boldsymbol{u}(t, \boldsymbol{x}_0, \boldsymbol{\mu}) - \boldsymbol{u}(t, \boldsymbol{y}_0, \boldsymbol{\mu})\| \leq \|\boldsymbol{x}_0 - \boldsymbol{y}_0\| e^{L(t - t_0)}. \tag{2}$$

*where $L$ is the uniform Lipschitz constant.*

The above theorem states the continuous dependence on the initial value, while its differentiality is given as follows.

**Corollary 1.** *If the function $\boldsymbol{f}(t, \boldsymbol{u}, \boldsymbol{\mu})$ is continuously differentiable with respect to $\boldsymbol{u}$, then its solution $\boldsymbol{u} = \boldsymbol{u}(t, \boldsymbol{u}_0, \boldsymbol{\mu})$ is also continuously differentiable with respect to initial values $\boldsymbol{u}_0$.*

**Remark 1.** *Regarding the statement of Corollary 1, we are straightforward to show*

$$\frac{d\boldsymbol{z}}{dt} = \frac{\partial \boldsymbol{f}}{\partial \boldsymbol{u}} \boldsymbol{z},$$
$$\boldsymbol{z}(t_0, \boldsymbol{\mu}) = 1, \tag{3}$$

where $\boldsymbol{z} = \partial \boldsymbol{u}/\partial \boldsymbol{u}_0$ denotes the continuous partial derivatives of solution $\boldsymbol{u}$ with respect to initial values $\boldsymbol{u}_0$.

Next, suppose that parameters $\boldsymbol{\mu}$ vary while initial values are fixed. Then we can obtain quite similar conclusions on the continuous dependence of the ODE solutions on the parameters, that is:

**Theorem 3** (continuous dependence on parameters). *For any $t_0 \in I$, $\boldsymbol{u}_0 \in W$ and $\boldsymbol{\mu}_0 \in V$, there exist constants $\alpha > 0$ and $\rho > 0$, such that when $|\boldsymbol{\mu} - \boldsymbol{\mu}_0| \leq \rho$, the solution of equation 1 with initial conditions $\boldsymbol{u}(t_0) = \boldsymbol{u}_0$ is defined on the interval $[t_0 - \alpha, t_0 + \alpha]$, and is a continuous function of both $t$ and $\boldsymbol{\mu}$.*

**Corollary 2.** *If the function $f(t, \boldsymbol{u}, \boldsymbol{\mu})$ has continuous partial derivatives with respect to variables $\boldsymbol{u}$ and $\boldsymbol{\mu}$, then the solution of equation 1 with the initial condition $\boldsymbol{u}(t_0) = \boldsymbol{u}_0$ is continuously differentiable with respect to $\boldsymbol{\mu}$.*

**Remark 2.** *The proof of Corollary 2 not only shows that $\partial \boldsymbol{u} / \partial \boldsymbol{\mu}$ exists and is continuous, but also satisfies the following differential equation:*

$$\frac{d\boldsymbol{z}}{dt} = \frac{\partial \boldsymbol{f}}{\partial \boldsymbol{u}} \boldsymbol{z} + \frac{\partial \boldsymbol{f}}{\partial \boldsymbol{\mu}},$$
$$\boldsymbol{z}(t_0, \boldsymbol{\mu}) = 0, \tag{4}$$

by introducing a new variable $\boldsymbol{z} = \partial \boldsymbol{u} / \partial \boldsymbol{\mu}$. Above formula together with the one in Remark 1, provide the basic theoretical foundation for the problems we consider in the next section.

## 2.2 PINN WITH CONTINUOUS DEPENDENCE (CD-PINN)

When applying PINN as well as its most variants to ODEs, most of them lack sufficient ability to maintain their outstanding performance on new parameters or initial values, which are dramatically different from the original ones used for training. Towards this key issue, our goal is to improve the generalization ability of PINN for solving various differential equations in a significant way. Meanwhile, the flexibility, efficiency, and accuracy of PINN should be kept as much as possible. In this way, we could easily solve differential equations under a diverse set of initial conditions and parameters, by using PINN after a small amount of training. This is particularly useful in applications requiring extensive predictions and real-time feedback, as it significantly enhances the system's robustness and efficiency while reducing the complexity of model maintenance.

Here we propose a variant of PINN, called cd-PINN (PINN with continuous dependence). We still adopt a deep neural network to approximate the ODE solution $\hat{\boldsymbol{u}}(t, \boldsymbol{u}_0, \boldsymbol{\mu})$. In the vanilla PINN, the loss function consists of two components: a supervised loss from data measurement of $\boldsymbol{u}$, which helps to stabilize the training procedure, and an unsupervised residual loss, which incorporates the physical information about the differential equations. Due to the regularity assumption on the function $f$, we know that the ODE solution $\boldsymbol{u}$ is continuously differentiable with respect to both parameters $\boldsymbol{\mu}$ and initial values $\boldsymbol{u}_0$. As a consequence, we will incorporate this valuable information on the continuous dependence into the loss function based on formulas in Remarks 1 and 2.

To sum up, the loss function of our cd-PINN is given by

$$\mathcal{L}(\boldsymbol{\theta}) = \lambda_{data} \mathcal{L}_{data} + \lambda_{res} \mathcal{L}_{res} + \lambda_{cd} \mathcal{L}_{cd}, \tag{5}$$

where

$$\mathcal{L}_{data} = \frac{1}{N_{data}} \sum_{i=1}^{N_{data}} \|\hat{\boldsymbol{u}}(t_i, \boldsymbol{u}_{0_i}, \boldsymbol{\mu}_i) - \boldsymbol{u}(t_i, \boldsymbol{u}_{0_i}, \boldsymbol{\mu}_i)\|_2^2$$

$$\mathcal{L}_{res} = \frac{1}{N_t} \sum_{i=1}^{N_t} \left\| \frac{d\hat{\boldsymbol{u}}}{dt}(t_i, \boldsymbol{u}_{0_i}, \boldsymbol{\mu}_i) - \boldsymbol{f}(t_i, \hat{\boldsymbol{u}}_i, \boldsymbol{\mu}_i) \right\|_2^2 + \frac{1}{N_0} \sum_{j=1}^{N_0} \|\hat{\boldsymbol{u}}(t_0, \boldsymbol{u}_{0_j}, \boldsymbol{\mu}_j) - \boldsymbol{u}_{0_j}\|_2^2$$

$$\mathcal{L}_{cd} = \frac{1}{N_t} \sum_{i=1}^{N_t} \left\| \left( \frac{\partial^2 \hat{\boldsymbol{u}}}{\partial \boldsymbol{\mu} \partial t} - \frac{\partial \boldsymbol{f}}{\partial \boldsymbol{u}} \frac{\partial \hat{\boldsymbol{u}}}{\partial \boldsymbol{\mu}} - \frac{\partial \boldsymbol{f}}{\partial \boldsymbol{\mu}} \right)(t_i, \boldsymbol{u}_{0_i}, \boldsymbol{\mu}_i) \right\|_2^2 + \frac{1}{N_0} \sum_{j=1}^{N_0} \left\| \frac{\partial \hat{\boldsymbol{u}}}{\partial \boldsymbol{\mu}}(t_0, \boldsymbol{u}_{0_j}, \boldsymbol{\mu}_j) \right\|_2^2 \tag{6}$$

$$+ \frac{1}{N_t} \sum_{i=1}^{N_t} \left\| \left( \frac{\partial^2 \hat{\boldsymbol{u}}}{\partial \boldsymbol{u}_0 \partial t} - \frac{\partial \boldsymbol{f}}{\partial \boldsymbol{u}} \frac{\partial \hat{\boldsymbol{u}}}{\partial \boldsymbol{u}_0} \right)(t_i, \boldsymbol{u}_{0_i}, \boldsymbol{\mu}_i) \right\|_2^2 + \frac{1}{N_0} \sum_{j=1}^{N_0} \left\| \frac{\partial \hat{\boldsymbol{u}}}{\partial \boldsymbol{u}_0}(t_0, \boldsymbol{u}_{0_j}, \boldsymbol{\mu}_j) - 1 \right\|_2^2.$$

Here $\{(t_i, \boldsymbol{u}_{0_i}, \boldsymbol{\mu}_i)\}$ represent those points sampled from the entire domain, while points $\{(t_0, \boldsymbol{u}_{0_j}, \boldsymbol{\mu}_j)\}$ are restricted to the initial time point $t_0$. The total number of sampled points is given by $N_t$ and $N_0$ separately. The weights $\lambda_{data}, \lambda_f$, and $\lambda_c$ are used to balance the interplay between the three loss terms. The subscript $\boldsymbol{\theta}$ is omitted for simplicity. Our non-trivial modification by introducing the loss function $\mathcal{L}_{cd}$ (named as the continuity loss)

helps to improve the generalization ability of PINN to a considerable extent, which will be addressed through fruitful examples in detail later.

In the literature, there are works which incorporate derivative (or smoothness) loss into the loss function. For example, Virmaux & Scaman (2018) proposed methods to constrain the Lipschitz constants of neural networks, while Song et al. (2023) developed networks with adaptive Lipschitz constants to achieve smoother outputs in reinforcement learning. These works mainly focus on improving the smoothness and robustness of neural networks, which make a clear distinction from our method.

# 3 Numerical Results

## 3.1 The Logistic Model

As the first example, we select the Logistic model, a classical ordinary differential equation first published by Pierre Verhulst to describe the population growth regulated by the carrying capacity due to resource limits. It reads

$$\frac{du}{dt} = ru\left(1 - \frac{u}{u_{max}}\right),\tag{7}$$

where $u = u(t)$ represents the population size, $u_{max}$ denotes the carrying capacity, and $r \in \mathbb{R}^+$ is the rate of maximum population growth. By using the separation of variables, we can find its general solution as $u^*(t) = u(t)/u_{max} = [1 + e^{-rt}(u_{max}/u_0 - 1)]^{-1}$, where $u_0$ is the initial population size. It is straightforward to verify that this solution is continuously differentiable with respect to the growth rate $r$ and the initial value $u_0$.

This simple equation could be easily solved by most state-of-the-art deep learning algorithms. For instance, by using PINN the relative error of solutions could be readily minimized below $10^{-3} - 10^{-4}$, although this good performance is limited to fixed growth rate $r$ and initial population size $u_0$. If we directly transfer the PINN model to a new growth rate or initial value without further fine-tuning, the predicted solution will significantly deviate from the true value (see Figure 1(a)). This fact clearly reveals that the vanilla PINN lacks sufficient ability in generalization.

In contrast, by taking the information on the continuous dependence of solutions to the Logistic equation on both the growth rate $r$ and initial population size $u_0$ into consideration (the carrying capacity $u_{max}$ is fixed), our cd-PINN can achieve a good agreement between the predicted population size and its true value at any time points. Most astonishingly, by only using single set of training data with respect to specified $r = 1$ and $u_0 = 0.3$ as marked by the red star in Figure 1(b), the absolute errors could be maintained below $10^{-1}$ (and in most regions below $10^{-3}$) over the entire region for $u_0^* = u_0/u_{max} \in [0.01, 1.0]$ and $r \in [0.1, 10.0]$. Moreover, we compare the absolute error of cd-PINN with that of PINN at each point in the parameter space. As illustrated in Figure 1(b), except for several tiny domains around the training data point, the accuracy of cd-PINN is significantly higher than PINN.

To uncover why cd-PINN has such a nice performance on generalization in the current study, we make a comparison on the vector fields of $\partial u/\partial r$ and $\partial u/\partial u_0$ given by PINN and cd-PINN in Figure 1(d). It can be observed that by imposing constraints not only on the solution $u(t)$ but also on its partial derivatives $\partial u/\partial r$ and $\partial u/\partial u_0$, cd-PINN has successfully reproduced the correct vector fields. Contrarily, the vanilla PINN makes wrong predictions, especially at the top-left corner of Figure 1(d).

In this task, the training data consists of 20 real data points corresponding to the solution of $u_0 = 0.3$, $r = 1.0$, along with $2^{14}$ residual data points, i.e. $N_t = N_0 = 2^{14}$. To approximate the continuous solution, we utilize a fully connected neural network with 6 hidden layers, each containing 64 neurons. And $Tanh$ is employed as the activation function. During the training procedure, the Adam optimizer is first applied for 50000 epochs, followed by an additional optimization using the LBFGS optimizer. The learning rate for the Adam optimizer is set to 0.001, while the learning rate for the LBFGS optimizer is set to 0.1.

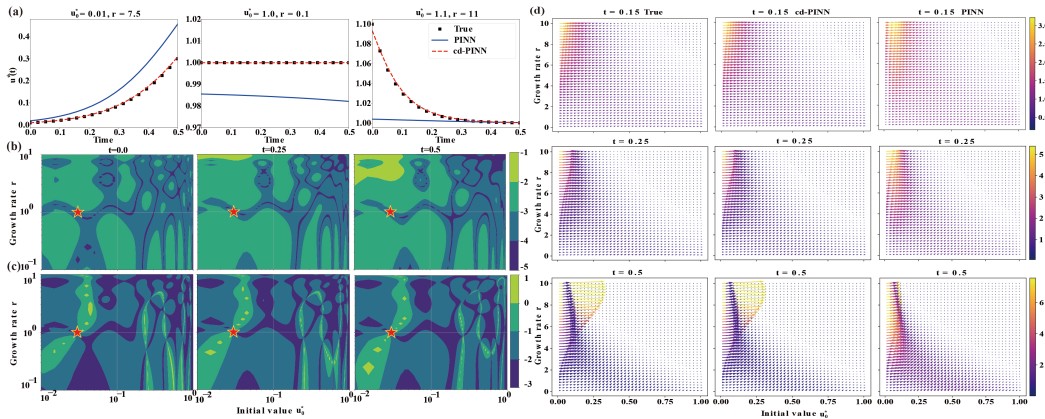

Figure 1: Comparison on the performance of cd-PINN v.s. PINN on the Logistic equation. Panel (a) depicts the exact solution, the predicted solutions of cd-PINN and PINN under specific initial value $u_0$ and parameter $r$. (b) illustrates the logarithms of absolute errors of cd-PINN over a wide range of the growth rate $r$ and normalized initial population size $u_0^*$. (c) displays the difference in the logarithm of absolute errors between cd-PINN and PINN (d) shows the vector fields of $\partial u / \partial r$ (upper row) and $\partial u / \partial u_0$ (lower row).

### 3.2 The Lotka-Volterra Model

The Lotka-Volterra model (LV model) consists of two coupled ordinary differential equations, representing the population changes of the predator and prey, respectively.

$$
\begin{aligned}
\frac{dX}{dt} &= c_{11}X + c_{12}XY + c_{13}X^2, \\
\frac{dY}{dt} &= c_{21}Y + c_{22}XY + c_{23}Y^2.
\end{aligned}
\tag{8}
$$

The initial values are $X(t=0) = X_0, Y(t=0) = Y_0$. We designed three scenarios for this example. In the first scenario, the system has an unstable fixed point at $(0,0)$, two stable fixed point at $(-\frac{c_{11}}{c_{13}}, 0)$ and $(-\frac{c_{12}c_{21} - c_{11}c_{23}}{c_{12}c_{21} - c_{13}c_{23}}, -\frac{c_{11}c_{22} - c_{13}c_{21}}{c_{12}c_{22} - c_{13}c_{23}})$. The primary goal of this scenario is to test whether the model can learn the correct solution when the initial value range spans the attraction domains of two different stable points.

We uniformly select 1600 groups of initial values of $(X_0, Y_0)$ during the interval $[0.1, 10] \times [0.1, 10]$ to generate the test data. Meanwhile, the real data consists of 20 points corresponding to the solution with initial values $X_0 = 8.0, Y_0 = 1.0$, and 20 points corresponding to the solution with initial values $X_0 = 5.0, Y_0 = 0.0$. The training data set includes the real data and $N_t = N_0 = 2^{14}$ residual data points.

As clearly seen in Figure 2(d), cd-PINN exhibits a quite promising generalization ability over a wide region of $(X_0, Y_0)$ with absolute errors smaller than $10^{-2}$, except for few points near domain boundaries. At the same time, the MSE of cd-PINN on the test data drops much faster than that of vanilla PINN with respect to training iterations (see Figure 2(b)). Furthermore, as highlighted through the zoomed-in plot in Figure 2(e), the phase plane predicted by PINN is inconsistent with the explicit one around the fixed point $(\frac{c_{11}}{c_{13}}, 0)$, which leads to an intrinsic qualitative difference from our cd-PINN. Details on the other two scenarios could be found in Appendix B.2.

Furthermore, our numerical simulations reveal that even for fixed initial values or parameters, the accuracy and convergence rate of cd-PINN are usually much better than PINN (see Appendix B.5.). Meanwhile, for unseen initial values and parameters, like data points beyond the training set, the cd-PINN also shows a satisfactory performance (data not shown), demonstrating a major strength of cd-PINN that it can indeed generalize to genuinely novel

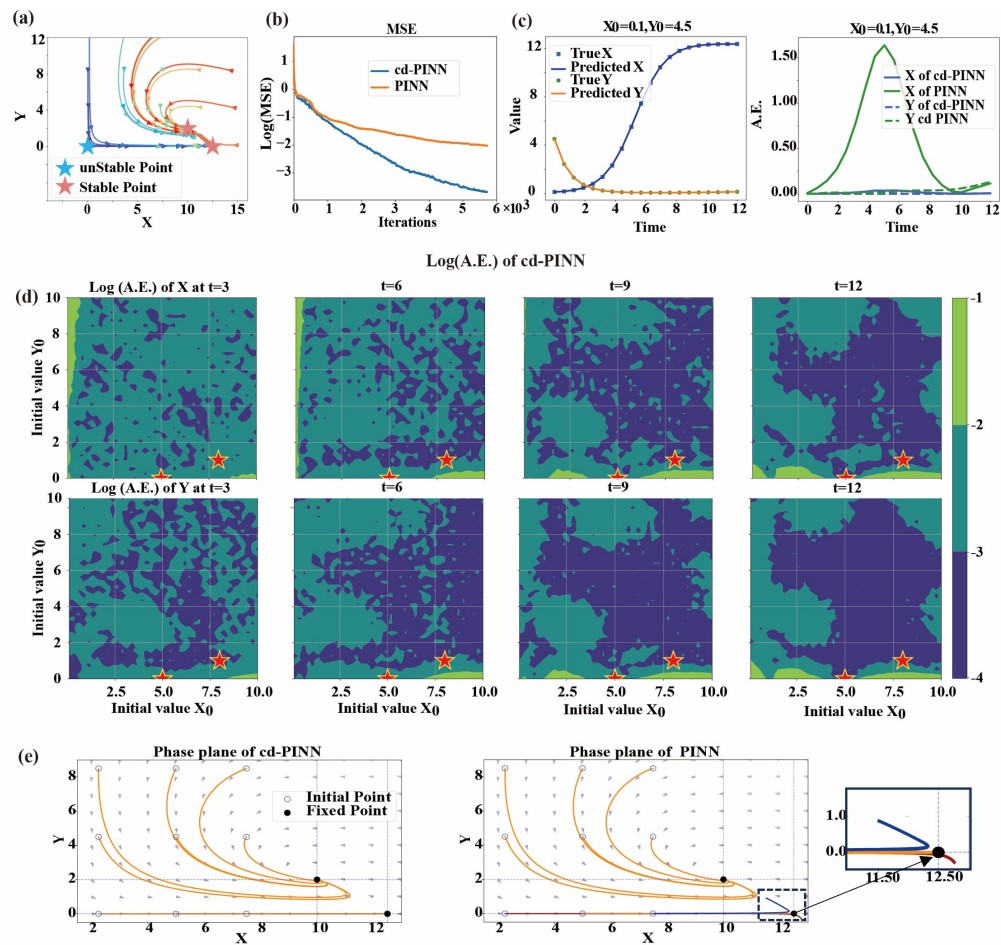

Figure 2: Results of LV equations under scenario one. (a) The phase plane of LV equations. (b) The MSE of test data for both PINN and cd-PINN. (c) The predicted solutions of cd-PINNs are compared with their exact solutions with respect to specific initial conditions (left panel), alongside comparisons on the absolute errors of cd-PINN and PINN (right panel). (d) The logarithms of absolute errors between the predicted solutions $X$ (upper row) and $Y$ (lower row) of cd-PINN and their respective true values. (e) Comparison on the predicted phase planes of cd-PINN and PINN. The predicted domain of PINN, which is inconsistent with the explicit result, is highlighted through the zoomed-in plot.

scenarios. We contribute these improvements to the inclusion of additional mathematical constraints on continuous dependence.

## 3.3 DAMPED HARMONIC OSCILLATOR

The damped harmonic oscillator is a system that moves back and forth around its equilibrium position in the presence of spring force and frictions. Mathematically, it is described by a second-order differential equation

$$\frac{d^2x}{dt^2} + 2\zeta\omega_0\frac{dx}{dt} + \omega_0^2 x = 0 \tag{9}$$

where $x$ is the displacement, $\zeta$ is the damping ratio, and $\omega_0$ is the intrinsic frequency, which is related to the spring constant in physics. The initial conditions are typically defined as $u(0) = u_0, \frac{du}{dt}(0) = v_0$, where $u_0$ is the initial displacement and $v_0$ is the initial velocity.

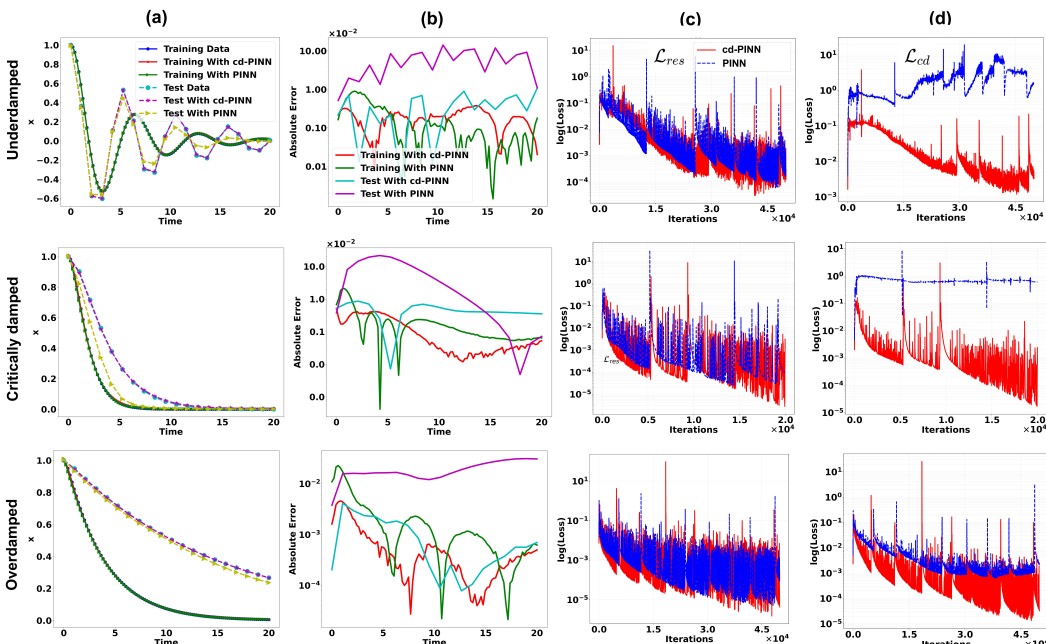

Figure 3: Comparison on the performance of cd-PINN v.s. PINN for the damped harmonic oscillator. Time evolution of (a) predicted solutions compared to training and test data, as well as (b) their absolute errors for under-, critically, and over-damped cases. (c) Residual loss $L_{res}$ and continuity loss $L_{cd}$ for cd-PINN and PINN over iterations.

Based on the damping ratio, the system can be classified into three distinguished cases when the damping ratio is relatively weak ( $0 < \zeta < 1$), the system keeps oscillating with gradually decreasing amplitude for a very long time, which is known as the underdamped case. Contrarily, when the damping ratio is strong ( $\zeta > 1$), the system returns to its equilibrium position as quickly as possible with no oscillation, which is called the overdamped case. Between these two, we have the critically damped case ($\zeta = 1$).

In all three cases, the training set is focused on a single parameter configuration with 100 evenly sampled time points from 0 to 20. The test sets, in contrast, explore a much broader parameter space. For the underdamped case, we take $\zeta = 0.2$ and $\omega_0 = 1.0$ for training, while the test set is spanned over $\zeta \in [0.1, 0.9]$ and $\omega_0 \in [0.5, 5.0]$. The overdamped case adopts $\zeta = 2.0$ and $\omega_0 = 1.0$ for training, with test parameters $\zeta \in [1.1, 5.0]$ and $\omega_0 \in [0.5, 5.0]$. Both cases result in 1600 distinct parameter combinations for testing. The critically damped case ($\zeta = 1.0$) mainly examines the influence of intrinsic frequency variations on the test set, with $\omega_0 \in [0.5, 5.0]$ taking 40 discrete points.

Our numerical experiments clearly reveal the high accuracy of cd-PINN in fitting both training and test data in all three damping cases, particularly in the underdamped and critically damped cases, as illustrated in Figure 3(a-b). In contrast, the vanilla PINN without considering continuous dependence of solutions exhibits much larger deviations from the test data. To gain a deep insight into the outstanding performance of cd-PINN, we make a direct comparison on the residual loss $\mathcal{L}_{res}$ and the continuity loss $\mathcal{L}_{cd}$ between cd-PINN and PINN. In Figure 3(c-d), we can see that the convergence rates of cd-PINN and PINN are comparable on the residual loss $\mathcal{L}_{res}$, while the continuity loss of cd-PINN converges much faster and is also much lower than that of PINN. This observation emphasizes the efficacy of integrating constraints on continuous dependence, resulting in a model with largely improved generalization capabilities.

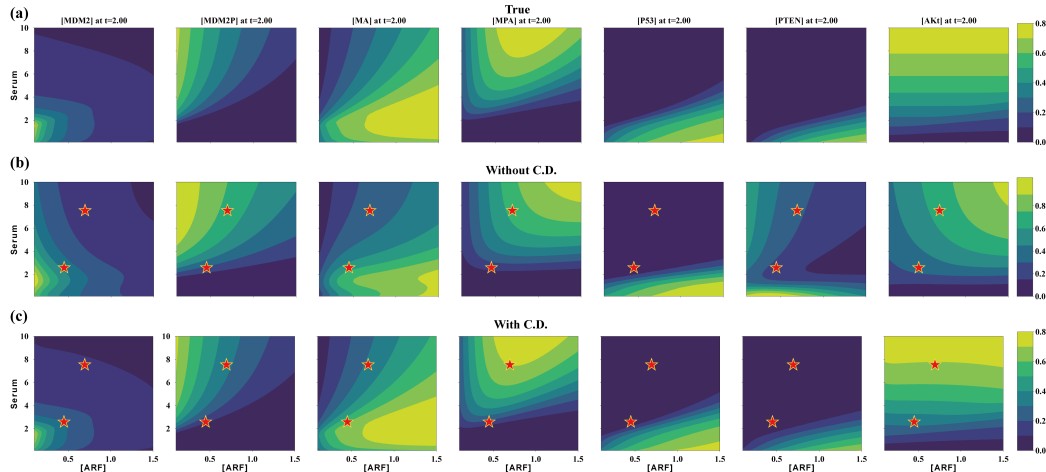

Figure 4: Phase diagram for the solutions of p53 activation model at time $t = 2$. The expression levels for seven genes are calculated by (a) the ODE solver, (b) PINN without C.D. constraints, and ($c$) cd-PINN separately in comparison.

### 3.4 A Multiscale Model for P53 Activation

In order to test whether our method is applicable to more complicated cases, here we look into a multiscale model for p53 activation, which is a key gene closely related to cancer development Tian et al. (2017). This model is composed of seven coupled ordinary differential functions, with the initial values (ranging from 0.005 to 5) and model parameters (ranging from 0.05 to 50) directly taken from the cited paper (see Appendix B.4. for the model and parameters).

In this task, we expect that the cd-PINN can correctly learn the solutions to the above model from time $t = 0$ to $t = 5$ with respect to arbitrary inputs of $[S] \in [0.1, 10.0]$ and $[ARF] \in [0.1, 1.5]$. For this purpose, we uniformly select $41 \times 41$ groups of $[S]$ and $[ARF]$ during the interval $[0.1, 10.0] \times [0.1, 1.5]$ to generate the test data. Meanwhile, the real data consists of 51 points corresponding to the solution with $[S] = 2.575$, $[ARF] = 0.45$ and 51 points corresponding to the solution with $[S] = 7.525$, $[ARF] = 0.695$. The training data set includes the real data and $N_t = N_0 = 2^{13}$ residual data points.

The final predictions of cd-PINN on this example are summarized in Figure 4, whose MSE is $3.32 \times 10^{-4}$, two order lower than the MSE of the model without C.D. constraints ($5.38 \times 10^{-2}$). Therefore, it can be concluded that our cd-PINN is capable for handling more challenging situations.

| System type | | PINN | | | cd-PINN | | |
|---|---|---|---|---|---|---|---|
| | | Time($s$) | NRMSE | MSE | Time($s$) | NRMSE | MSE |
| | Logistic | $2,158$ | $1.17 \times 10^{-2}$ | $7.49 \times 10^{-3}$ | $2,751$ | $9.06 \times 10^{-4}$ | $4.48 \times 10^{-5}$ |
| LV | Scenery 1 | $414$ | $1.39 \times 10^{-2}$ | $9.24 \times 10^{-3}$ | $837$ | $8.12 \times 10^{-4}$ | $3.90 \times 10^{-5}$ |
| | Scenery 2 | $5,195$ | $6.97 \times 10^{-3}$ | $4.04 \times 10^{-4}$ | $7,088$ | $5.63 \times 10^{-4}$ | $3.12 \times 10^{-6}$ |
| | Scenery 3 | $1,090$ | $4.78 \times 10^{-3}$ | $2.51 \times 10^{-3}$ | $1,360$ | $3.22 \times 10^{-4}$ | $1.14 \times 10^{-5}$ |
| OS | Underdamped | $2,140$ | $5.38 \times 10^{-1}$ | $1.14 \times 10^{-3}$ | $3,999$ | $3.35 \times 10^{-1}$ | $4.43 \times 10^{-4}$ |
| | Critical | $1,000$ | $3.14 \times 10^{-1}$ | $3.83 \times 10^{-4}$ | $1,694$ | $7.46 \times 10^{-2}$ | $2.16 \times 10^{-5}$ |
| | Overdamped | $2,154$ | $6.12 \times 10^{-2}$ | $4.23 \times 10^{-5}$ | $4,040$ | $4.25 \times 10^{-2}$ | $2.04 \times 10^{-5}$ |
| | p53 activation | $4,017$ | $2.40 \times 10^{-1}$ | $5.38 \times 10^{-2}$ | $7,193$ | $1.88 \times 10^{-2}$ | $3.32 \times 10^{-4}$ |

Table 1: Summary on the training time and accuracy of cd-PINN v.s. PINN.

## 4 Conclusion and Discussion

Previous deep-learning-based algorithms have achieved fantastic results in various fields, though they still face with big challenges in generalization beyond the training data. Moreover, when applied to complex systems, like partial differential equations, where solutions exhibit sensitivity to initial conditions or model parameters, these weaknesses become more obvious.

To enhance the generalization capability of PINN, in this study we propose a novel approach (cd-PINN) by incorporating additional information on the continuous dependence of solutions on the parameters and initial values. Through the Logistic model, Lotka-Volterra equations, damped harmonic oscillators and a multiscale model for p53 activation, the significant advantages of our cd-PINN over the vanilla PINN have been clearly demonstrated, which are summarized as follows.

**Generalization and accuracy.** Incorporating the continuous dependence information into the loss function enables cd-PINN to effectively learn the fundamental mapping between parameters/initial values and solutions. In all numerical experiments, our cd-PINN shows a comparable accuracy to the vanilla PINN on the training data. More importantly, our cd-PINN could maintain its promising performance on new configurations, which are far away from the training data in the parameter space.

**Universality and robustness.** From the simple Logistic model to complex LV dynamics and oscillatory systems, our cd-PINN demonstrates its universal applicability. Moreover, it is observed that either the stability of fixed points or their respective attraction domains have limited impacts on the solution's accuracy, implying the robustness of cd-PINN against the underlying dynamics and the location of training data.

**Efficiency and no-fine-tuning.** The inclusion of the continuity loss does not apparently increase the computational cost of cd-PINN, whose training time is still comparable to the vanilla PINN. Meanwhile, cd-PINN has no demand for retraining or fine-tuning when facing with new parameters or initial conditions, in contrast to meta-PINN.

In the literature, Neural ODEs (Chen et al. (2018)) are another prominent approach for learning solutions to ordinary differential equations. With respect to the same data set and evaluation metrics, we find that the Neural ODE model could effectively capture the system dynamics at the training data points, but shows a much poorer generalization ability than cd-PINN under the testing scenarios (see Appendix B.6). In addition, the Neural ODE model takes a much longer training time due to the explicit implementation of temporal integration steps, whereas PINN and cd-PINN are more computationally efficient.

In the current paper, we restrict our study to ordinary differential equations for clarity. Obviously, the same approach is applicable to partial differential equations too, e.g. the viscosity in Burgers equation or the Reynolds number in Navier-Stokes equations. However, it should be noted that the PDE cases are far more complicated in general. For example, in many cases the parameter dependence of PDEs may be continuous but not necessarily differentiable. This subtle distinction is crucial for certain contexts, such as the shock structures in hyperbolic conservation laws(Evans (2022)). Under these situations, we need to turn to more general conditions, like the Rankine-Hugoniot jump condition, to determine the exact locations where the shock structure arises. The related work is ongoing. Furthermore, we would like to explore the advantages and limitations of cd-PINN in high-dimensional and multiscale systems, especially for real-world problems.

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
