# Appendix:

# Incorporating continuous dependence implies better generalization ability

## Appendix A  Physics-Informed Neural Networks

With the rise of deep learning, many researchers focus on using deep neural networks to solve various differential equations. One notable achievement is the Physics-Informed Neural Networks (PINN) (Raissi et al., 2019). In this approach, a key step is to incorporate the governing differential equations into the loss function. This greatly reduces the requirement for high-quality data for training, making it suitable for solving high-dimensional forward problems.

In PINN, the unknown solution $\boldsymbol{u}(t, \boldsymbol{x}, \boldsymbol{\mu})$ is represented by a deep neural network $\hat{\boldsymbol{u}}_{\boldsymbol{\theta}}(t, \boldsymbol{x}, \boldsymbol{\mu})$, where $\boldsymbol{\mu}$ represents the parameters in the differential equations and $\boldsymbol{\theta}$ denotes all trainable parameters of the network, including both weights and biases. To be concrete, let $\mathcal{N}^Q : \mathbb{R}^{1+n+m} \to \mathbb{R}^n$ be a feed-forward neural network with $Q$ layers, where the $k$-th layer contains $N_k$ neurons ($N_0 = 1+n+m$ and $N_L = n$). The weight matrix and bias vector in the $k$-th layer ($1 \le k \le Q$) are denoted by $\boldsymbol{W}^k \in \mathbb{R}^{N_k \times N_{k-1}}$ and $\boldsymbol{b}^k \in \mathbb{R}^{N_k}$ respectively. Denote the input vector as $\boldsymbol{z} = (t, \boldsymbol{u}_0, \boldsymbol{\mu}) \in \mathbb{R}^{1+n+m}$, then the output vector at $k$-th layer is given by $\mathcal{N}^k(\boldsymbol{z})$. In particular, we have $\mathcal{N}^0(\boldsymbol{z}) = \boldsymbol{z}$. Consequently, the operation in the $(Q-1)$-hidden layers is given through

$$\mathcal{N}^k(\boldsymbol{z}) = \boldsymbol{W}^k \sigma(\mathcal{N}^{k-1}(\boldsymbol{z})) + \boldsymbol{b}^k \in \mathbb{R}^{N_k}, \; 2 \le k \le Q, \tag{1}$$

and $\mathcal{N}^1(\boldsymbol{z}) = \boldsymbol{W}^1 \boldsymbol{z} + \boldsymbol{b}^1$, where $\sigma$ denotes the activation function. In the last hidden layer, the identity function is taken as the activation function. Letting $\boldsymbol{\theta} = \left\{ \boldsymbol{W}^k, \boldsymbol{b}^k \right\}$ be the collection of all weights and biases in the feed-forward neural network, we can write the output of the neural network as

$$\hat{\boldsymbol{u}}(\boldsymbol{z}) = \hat{\boldsymbol{u}}_{\boldsymbol{\theta}}(t, \boldsymbol{u}_0, \boldsymbol{\mu}) = \mathcal{N}^Q(\boldsymbol{z}; \boldsymbol{\theta}), \tag{2}$$

where $\mathcal{N}^Q(\boldsymbol{z}; \boldsymbol{\theta})$ emphasizes the dependence of the neural network output $\mathcal{N}^Q(\boldsymbol{z})$ on $\boldsymbol{\theta}$.

The mapping between points in the spatiotemporal domain and the solution of the differential equations are optimized by minimizing the following composite loss function

$$\mathcal{L}(\boldsymbol{\theta}) = \lambda_{data}\mathcal{L}_{data} + \lambda_{res}\mathcal{L}_{res}, \tag{3}$$

where $\mathcal{L}_{data}$ represents the loss at those training data points, while $\mathcal{L}_{res}$ denotes the residual loss incorporating physical information about the differential equations. $\lambda_{data}$ and $\lambda_{res}$ are weights assigned to each loss component.

In this study, we make adaptation of the PINN model, so that it accepts not only the space-time coordinates $x, t$ as input, but also the parameters and initial values of the equations. Such changes enable the PINN model to have a certain generalization ability on new configurations.

# Appendix B    Additional Results on Numerical Experiments

## B.1    The Logistic model

The distinction between PINN and cd-PINN is further revealed by an analysis of their respective loss functions. Here we compare each component of the loss function of cd-PINN. As shown in Figure B1(a), although the data loss of PINN is slightly lower, cd-PINN achieves a better residual loss during training. Most importantly, the continuity loss of cd-PINN is about two orders of magnitude lower than that of PINN, which provides a clear demonstration on why cd-PINN owns such an outstanding generalization ability. Furthermore, the loss landscape of PINN with respect to two principal components of the deep neural network appears to be quite rugged and has plenty of local minimums, in great contrast to the smooth one of cd-PINN (see Figure B1(b)).

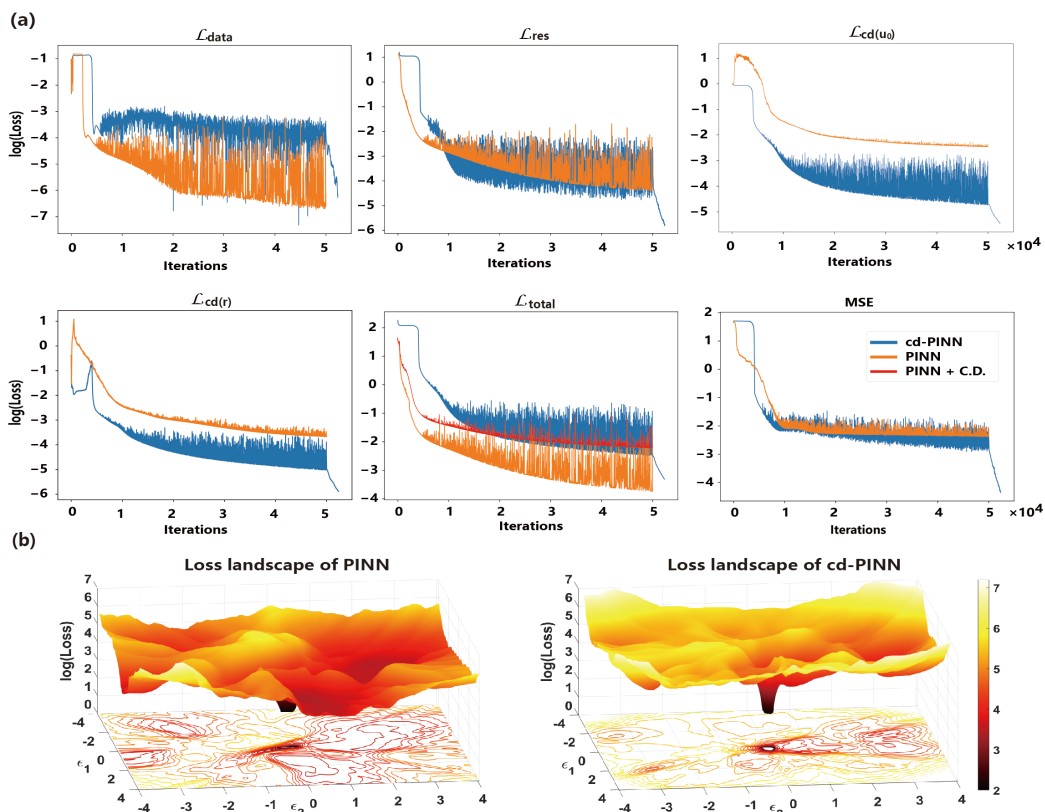

Figure B1: Loss function and loss landscape of cd-PINN v.s. PINN. (a) Trajectories of each component of the loss function and MSE for cd-PINN and PINN. (b) The loss landscape with respect to two principal components of the deep neural network after training.

## B.2 The Lotka-Volterra Model

The LV system has four equilibrium points $(0,0)$, $(-\frac{c_{11}}{c_{13}}, 0)$, $(0, -\frac{c_{21}}{c_{23}})$ and $(-\frac{c_{12}c_{21} - c_{11}c_{23}}{c_{12}c_{21} - c_{13}c_{23}}, -\frac{c_{11}c_{22} - c_{13}c_{21}}{c_{12}c_{22} - c_{13}c_{23}})$ under the condition $c_{12} \times c_{22} \neq c_{12} \times c_{23}, c_{13} \neq 0$ and $c_{23} \neq 0$. And the dynamical behaviors of the LV system around its equilibrium points $(X^*, Y^*)$ are fully characterized by the following Jacobian matrix:

$$J = \left[ \begin{array}{cc} c_{11} + 2c_{13}X + c_{12}Y & c_{12}X \\ c_{22}Y & c_{21} + 2c_{23}Y + c_{22}X \end{array} \right]_{(X^*, Y^*)}. \tag{4}$$

Specifically, when $c_{11} > 0$ and $c_{21} < 0$, the fixed point $(-\frac{c_{11}}{c_{13}}, 0)$ is stable. When $c_{11} < 0$, $c_{21} > 0$, the fixed point $(0, -\frac{c_{21}}{c_{23}})$ is stable. When both $c_{11} < 0$ and $c_{21} < 0$, the fixed point $(0,0)$ is stable. Finally, the fixed point $(-\frac{c_{12}c_{21} - c_{11}c_{23}}{c_{12}c_{21} - c_{13}c_{23}}, -\frac{c_{11}c_{22} - c_{13}c_{21}}{c_{12}c_{22} - c_{13}c_{23}})$ is a central point, which means that the orbits near it are periodic. Table $B1$ summarizes the parameter settings, stable points, and their respective attraction domains for the three scenarios considered in the numerical experiment.

Table B1: Parameters setup, fixed points, and attraction domain for stable fixed points in the LV model. For simplicity, we set $\Omega = [0, \infty) \times [0, \infty) \backslash \{0\} \times \{0\}$, and $(0,0)$ is an unstable point.

| Scenario | $c_{11}$ | $c_{12}$ | $c_{13}$ | $c_{21}$ | $c_{22}$ | $c_{23}$ | Fixed points | Attraction domain |
|---|---|---|---|---|---|---|---|---|
| 1 | 1.0 | -0.1 | -0.08 | -1.0 | 0.1 | 0.0 | $(0, 0), (12.5, 0), (10, 2)$ | $(0, \infty) \times \{0\}, \Omega \backslash (0, \infty) \times \{0\}$ |
| 2 | 1.0 | 0.0 | -0.5 | -0.1 | -0.1 | 0.0 | $(0, 0), (2, 0)$ | $\Omega$ |
| 3 | 1.0 | -0.1 | 0.0 | -1.0 | 0.1 | 0.0 | $(0, 0), (10, 10)$ | central point |

In experiments, we typically conduct trials with few groups of initial values, or even just a single group of initial values. However, it is essential to understand the solutions corresponding to a broader range of initial values surrounding this group. In scenario two, we test the generalization ability of cd-PINN to arbitrary initial values $(X_0, Y_0)$ in a wide range with only one group of initial values for training (as marked by the red five-pointed star). Specifically, the attraction domain of the stable point $(-\frac{c_{11}}{c_{13}}, 0) = (2, 0)$ is $[0, \infty) \times [0, \infty) \backslash \{0\} \times \{0\}$, as shown in the phase trajectory in Figure B2 (a). For practice, we select the test interval as $[0.1, 10.0] \times [0.1, 10.0]$, from which 1600 groups of initial values is uniformly sampled to generate the test data set. The training data consists of 20 real data points corresponding to the solution for $X_0 = 5.0$, $Y_0 = 5.0$, and $2^{14}$ residual data points to calculate $loss_f$ and $loss_{cd}$. Details of the neural network architecture and training method are presented in Table C5.

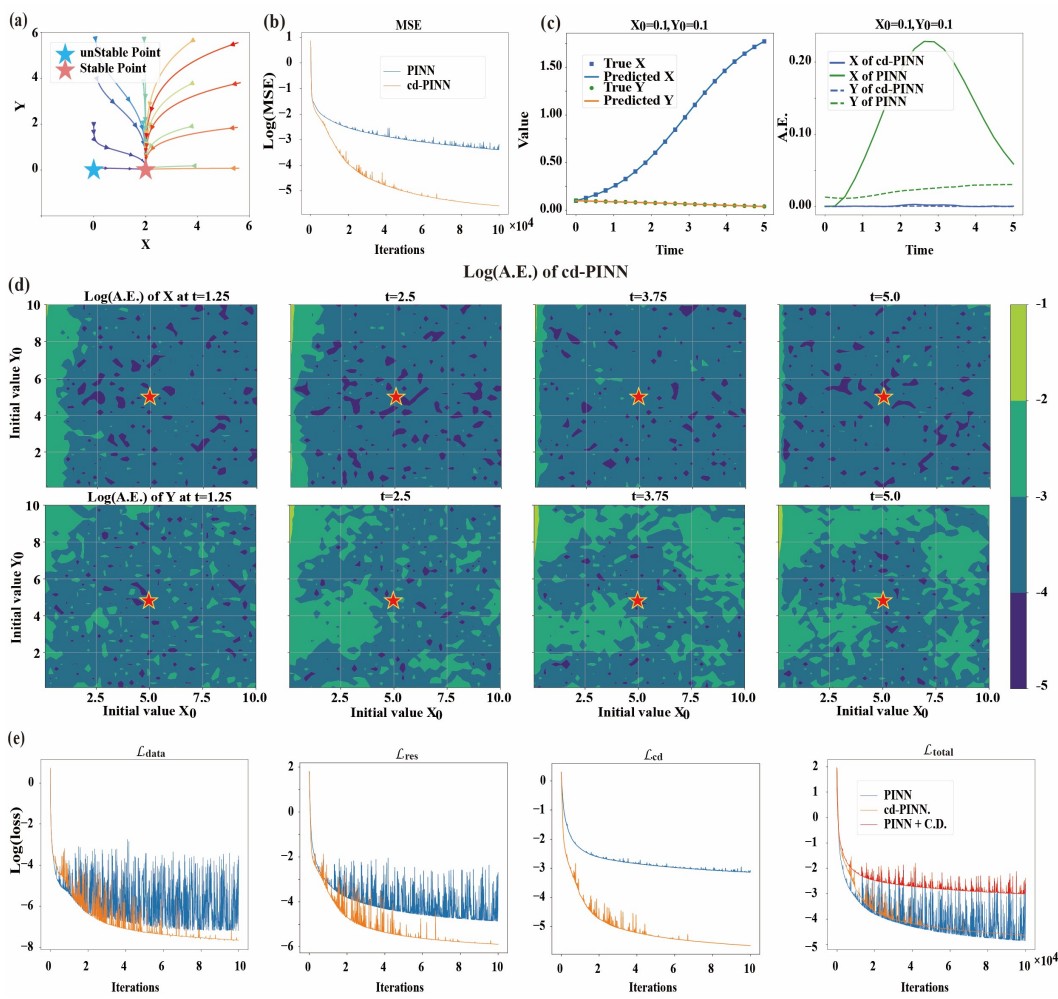

Figure B2: Results for LV equations under scenario two. (a) The phase plane of LV equations. (b) The MSE of test data for both PINN and cd-PINN during training iterations. (c) The predicted solutions of cd-PINNs are compared with their exact solutions with respect to different initial conditions (left panel), alongside comparisons on the absolute errors of cd-PINN and PINN (right panel). (d) The logarithm of absolute errors between the predicted solutions $X$ (upper row) and $Y$ (lower row) of cd-PINN and their respective true values. (e) Comparison on each component of the loss function for cd-PINN and PINN.

Scenario three is mainly intended to test whether cd-PINN can learn the correct solution when the system contains a central point and exhibits periodic orbits. The central point in this system, given the specified parameter set, is located at $(-\frac{c_{12}c_{21} - c_{11}c_{23}}{c_{12}c_{22} - c_{13}c_{23}}) = (10.0, 10.0)$. The corresponding phase trajectory diagram is shown in Figure B3(a). For practice, we uniformly sample 1600 groups of initial values from the domain $[5.0, 15.0] \times [5.0, 15.0]$ as the test set. Meanwhile, the training data set consists of 20 real data points corresponding to the solution with initial values $X_0 = 5.0, Y_0 = 5.0$, along with $2^{14}$ residual data points for calculating $loss_f$ and $loss_{cd}$.

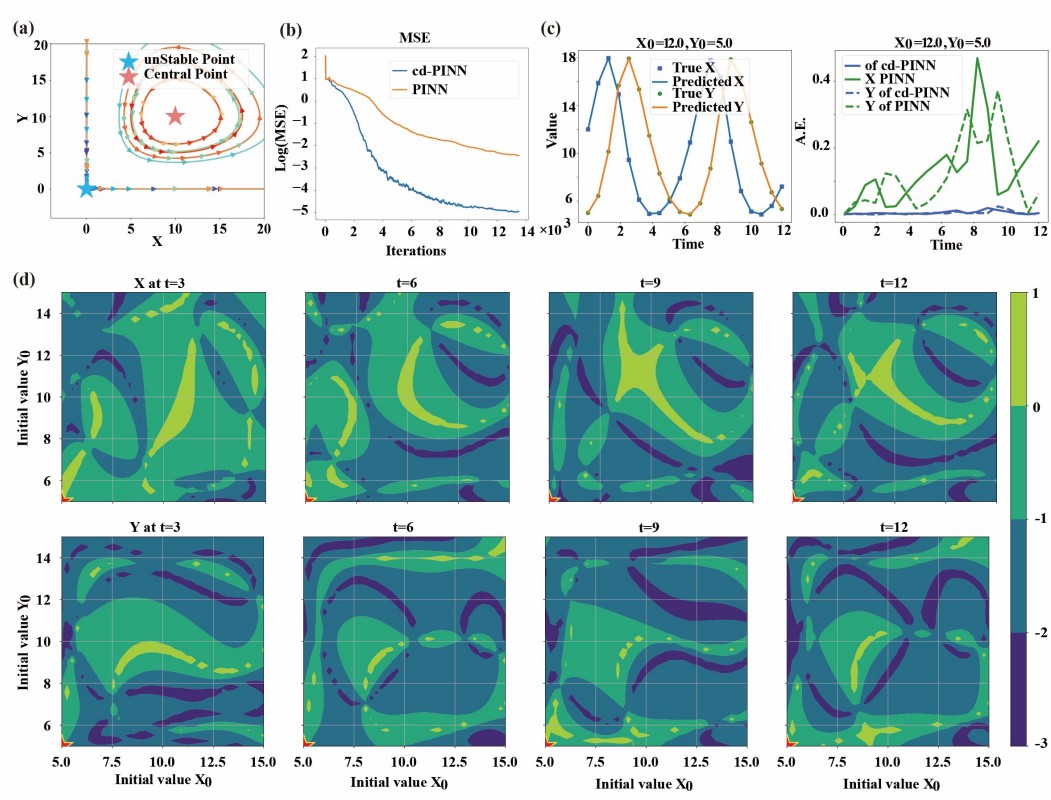

Figure B3: Result for LV equations under scenario three. (a) The phase plane of LV equations. (b) The MSE of test data for both PINN and cd-PINN during training iterations. (c) The predicted solutions of cd-PINNs are compared with their exact solutions with respect to different initial conditions (left panel), alongside comparisons on the absolute errors of cd-PINN and PINN (right panel). (d) Differences in the logarithm of absolute errors of cd-PINN and PINN for $X$ (upper row) and $Y$ (lower row).

### B.3 Damped Harmonic Oscillator

The harmonic oscillator is a system that moves back and forth around its equilibrium position. In the ideal case, without energy loss, the harmonic oscillator would maintain its motion eternally. However, the real-world system always experiences energy loss. Damping forces gradually reduce the amplitude of oscillations, eventually bringing the system back to its equilibrium state. Based on the damping ratio, we can distinguish three scenarios. The detailed setting of these three scenarios can be found in Table B2. In addition, the parameter setup for our numerical experiments on the damped harmonic oscillators is summarized in Table B3.

Table B2: Characteristics and solutions of damped harmonic oscillators under different conditions. Note in all three cases, $A$ and $B$ are constants determined by the initial conditions of the system.

| Cases | Characteristic Roots | Solutions |
|---|---|---|
| Underdamped $(0 < \zeta < 1)$ | $-\zeta\omega_0 \pm i\omega_0\sqrt{1-\zeta^2}$ | $x(t) = e^{-\zeta\omega_0 t}(A\cos(\omega_d t) + B\sin(\omega_d t))$ where $\omega_d = \omega_0\sqrt{1-\zeta^2}$ |
| Critically Damped $(\zeta = 1)$ | $-\omega_0$ | $x(t) = (A + Bt)e^{-\omega_0 t}$ |
| Overdamped $(\zeta > 1)$ | $-\zeta\omega_0 \pm \omega_0\sqrt{\zeta^2-1}$ | $x(t) = Ae^{(-p+q)t} + Be^{(-p-q)t}$ where $p = \zeta\omega_0$ and $q = \omega_0\sqrt{\zeta^2-1}$ |

Table B3: Parameter setup for the damped harmonic oscillator.

| Parameters | Underdamped | Critically Damped | Overdamped |
|---|---|---|---|
| | | Training Set | |
| Damping ratio ($\zeta$) | 0.2 (fixed) | 1.0 (fixed) | 2.0 (fixed) |
| $\omega_0$ | 1.0 (fixed) | 1.0 (fixed) | 1.0 (fixed) |
| Time range | $0 \sim 20$ | $0 \sim 20$ | $0 \sim 20$ |
| Number of time points | 100 | 100 | 100 |
| | | Test Set | |
| Damping ratio ($\zeta$) | $0.1 \sim 0.9$ | 1.0 (fixed) | $1.1 \sim 5.0$ |
| $\omega_0$ | | $0.5 \sim 5.0$ | |
| Time range | $0 \sim 20$ | $0 \sim 20$ | $0 \sim 20$ |
| Number of time points | 20 | 20 | 20 |
| Total configurations | 1600 (40 × 40) | 40 | 1600 (40 × 40) |
| Total data points | 32,000 (1600 × 20) | 800 (40 × 20) | 32,000 (1600 × 20) |

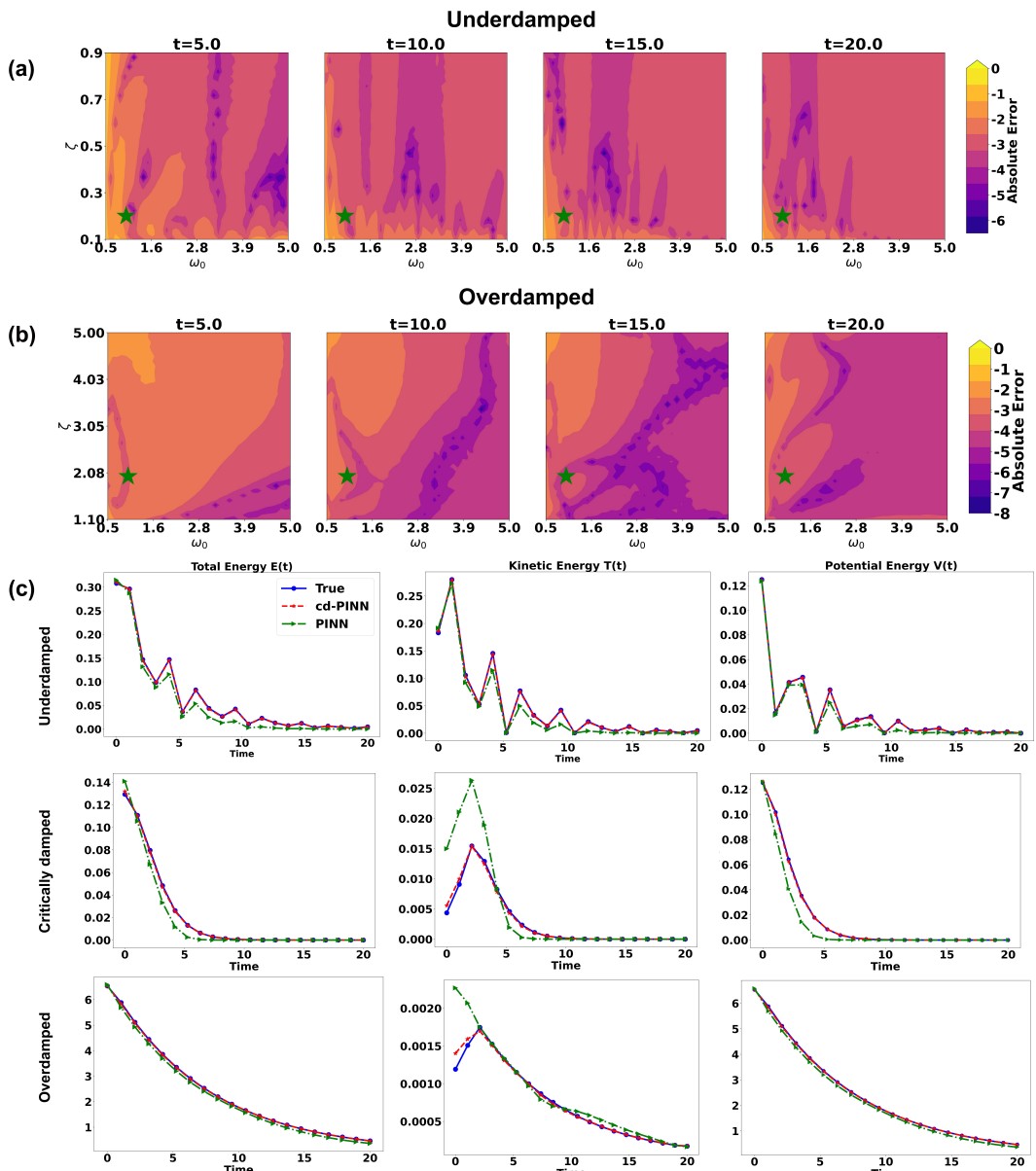

Figure B4: Absolute error distribution and energy evolution of cd-PINN for the damped harmonic oscillator. (a,b) show the distribution of the logarithm of absolute errors over the parameter space for the underdamped and overdamped systems, respectively, at various time points. The green five-pointed star indicates the training point. (c) The time evolution of total, kinetic and potential energies for the test data in the system of under-, critically, and over-damped oscillators separately.

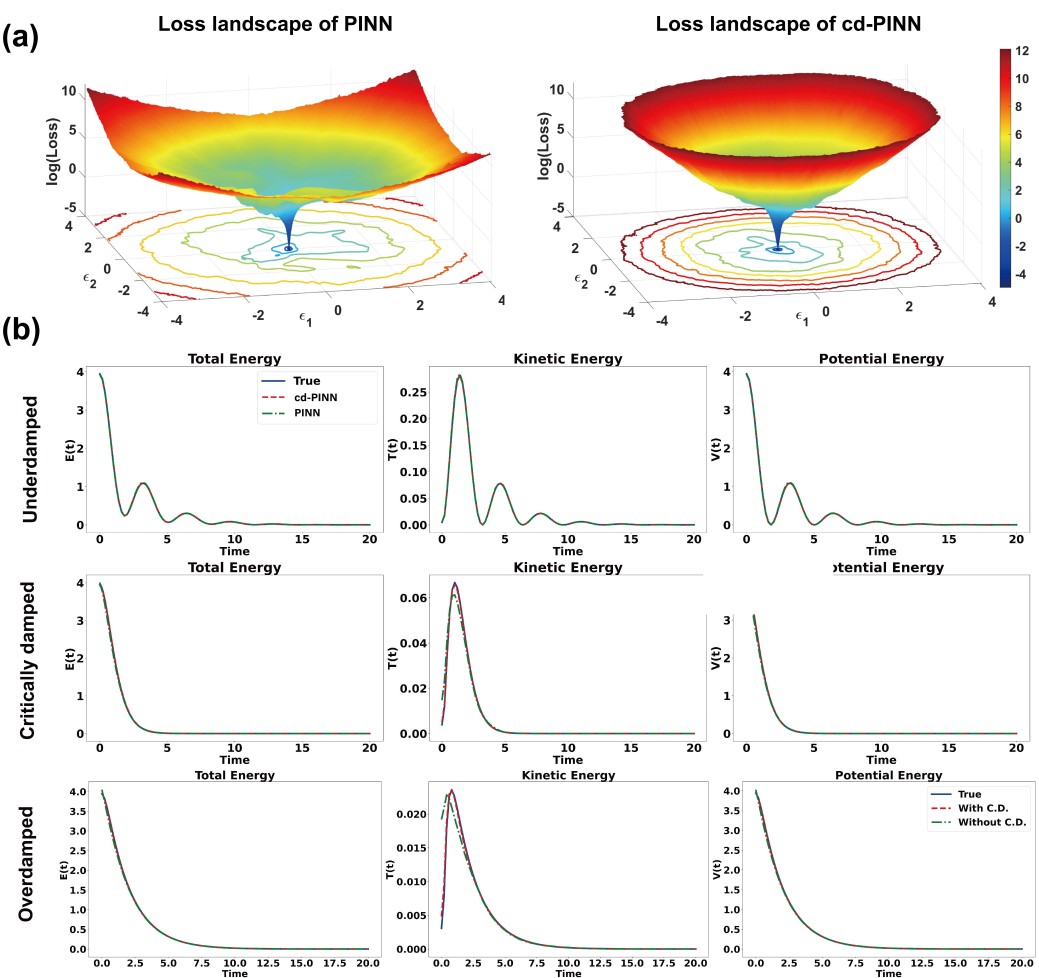

Figure B5: The loss landscape and energy evolution of cd-PINN for the damped harmonic oscillator. (a) displays the loss landscapes of PINN and cd-PINN for the overdamped oscillator. (b) shows the time evolution of total, kinetic and potential energies for the training data in the system of under-, critically, and over-damped oscillators separately.

During the study of underdamped and overdamped systems, we further analyze the performance of cd-PINN with respect to different model parameters $\zeta$ and $\omega_0$. Figure B4(a-b) illustrate the absolute error distributions of cd-PINN for both systems at four typical time points. Figure B4(c) and Figure B5(b) compare the time evolution of total energy, kinetic energy, and potential energy for the underdamped, critically damped, and overdamped systems. As can be seen, the cd-PINN model aligns more closely with the true energy, particularly in the underdamped and critically damped cases, where the model accurately tracks the true physical energy. In contrast, the PINN model shows significant deviations in energy prediction, especially in the evolution of kinetic and potential energies. A further calculation shows that the loss landscape of cd-PINN is relatively smoother and deeper than that of PINN, which in some way gives an explanation to the outstanding performance of cd-PINN.

### B.4   A Multiscale Model for P53 Activation

The p53 model considered in the main text is directly borrowed from the paper of Tian et al. (2017), which consists of seven coupled ordinary differential functions representing the expression levels of seven corresponding genes. It reads

$$
\frac{d[MDM2]}{dt} = \frac{k_{MD_S}[S]}{K_{MD_S} + [S]} + \frac{k_{M_p}[p53]^{n_1}}{K_{M_p}^{n_1} + [p53]^{n_1}} + D_{MA}[MA] + \frac{k_{Dp4}[MDM2_p]}{K_{Mp} + [MDM2_p]}
$$

$$
\qquad - k_{MA}[MDM2][ARF] - \frac{k_{p_M}[Akt][MDM2]}{K_{Akt_M} + [MDM2]} - d_{Mdm2}[MDM2],
$$

$$
\frac{d[MDM2_p]}{dt} = D_{MpA}[M_pA] + \frac{k_{p_M}[Akt][MDM2]}{K_{Akt_M} + [MDM2]} - k_{MpA}[MDM2_p][ARF]
$$

$$
\qquad - \frac{k_{Dp4}[MDM2_p]}{K_{Mp} + [MDM2_p]} - d_{Mp}[MDM2_p],
$$

$$
\frac{d[MA]}{dt} = k_{MA}[MDM2][AFR] - D_{MA}[MA] - d_{MA}[MA], \tag{5}
$$

$$
\frac{d[M_pA]}{dt} = k_{MpA}[MDM2_p][ARF] - D_{MpA}[M_pA] - d_{MpA}[M_pA],
$$

$$
\frac{d[p53]}{dt} = k_{p53} - \frac{k_{M53}[MDM2][p53]}{K_{M53} + [p53]} - \frac{k_{Mp53}[MDM2_p][p53]}{K_{Mp53} + [p53]} - d_{p53}[p53],
$$

$$
\frac{d[PTEN]}{dt} = k_{PTEN} + \frac{k_{P_p}[p53]^{n_2}}{K_{P_p}^{n_2} + [p53]^{n_2}} - d_{PTEN}[PTEN],
$$

$$
\frac{d[Akt]}{dt} = \frac{k_{A_S}[S]}{K_{A_S} + [S]} \frac{[Akt]_t - [Akt]}{K_0 + [Akt]_t - [Akt]} - \frac{k_{DP3}[Akt]}{K_{Akt} + [Akt]} - \frac{k_{A_p}[PTEN][Akt]}{K_{AP} + [Akt]}.
$$

The initial conditions are set as $([MDM2], [MDM2_p], [MA], [M_pA], [p53], [PTEN], [Akt]) = (0.2, 4.76, 0.038, 0.058, 0.006, 0.1, 0.78)\mu M$, while the reaction rate constants used for calculation are summarized as follows.

Table B4: Standard parameter values for the p53 activation model.

| Parameter | Value | Interpretation |
|---|---|---|
| $k_{MD_S}$ | 0.66 $\mu M/h$ | Rate constant of $MDM2$ expression induced by serum; it is nearly twice the degradation of $MDM2$ |
| $k_{M_p}$ | 0.33 $\mu M/h$ | Rate constant of p53-activation expression of $MDM2$ |
| $k_{DP4}$ | 12 $\mu M/h$ | Rate constant for $MDM2_p$ dephosphorylation |
| $k_{MA}$ | 43 $/(\mu M \cdot h)$ | Rate constant for MDM2/ARF association |
| $k_{P_M}$ | 56 $/h$ | Rate constant for $MDM2$ phosphorylation mediated by Akt |
| $k_{MpA}$ | 10 $/(\mu M \cdot h)$ | Rate constant for $MDM2_p/ARF$ association |
| $k_{p53}$ | 4.8 $\mu M/h$ | Basal rate constant of p53 expression, estimated at $0.005 \sim 0.2 \mu M/min$ |
| $k_{M53}$ | 5 $/h$ | Rate constant for $MDM2-$mediated $p53$ degradation, assumed to be slower than that mediated by phosphorylated MDM2 |
| $k_{Mp53}$ | 18 $/h$ | Rate constant for $MDM2_p-$ mediated $p53$ degradation, assumed to be $5-$ fold $d_{p53}$ |
| $k_{PTEN}$ | 0.05 $\mu M/h$ | Basal expression rate of $PTEN$ |
| $k_{P_p}$ | 0.7 $\mu M/h$ | Rate constant of $p53-$dependent synthesis of $PTEN$ |
| $k_{A_S}$ | 12.9 $\mu M/h$ | Rate constant for $Akt$ phosphorylation induced by growth factors |
| $k_{DP4}$ | 9.6 $\mu M/h$ | Rate constant for Akt dephosphorylation |
| $k_{A_P}$ | 30 $/h$ | Rate constant for $PTEN-$ induced $Akt$ dephosphorylation; referring to the dephosphorylation of $PIP3$ by $PTEN$ |
| $K_{Akt_M}$ | 0.5 $\mu M$ | Michaelis constant for Akt-mediated phosphorylation of $MDM2$ |
| $K_{MD_S}$ | 0.45% | Michaelis constant for $MDM2$ expression triggered by growth factors |
| $K_{M_p}$ | 0.5 $\mu M$ | Hill constant for $p53-$ induced expression of $MDM2$ |
| $K_{Mp}$ | 0.081 $\mu M$ | Michaelis constant for $MDM2_p$ dephosphorylation |
| $K_{M53}$ | 0.5 $\mu M$ | Assumed to be $5-$ fold that mediated by $MDM2_p$ |
| $K_{Mp53}$ | 0.1 $\mu M$ | Michaelis constant for $MDM2-$ mediated $p53$ degradation |
| $K_{P_p}$ | 1 $\mu M$ | Hill constant for $p53-$ induced expression of $MDM2$ |
| $K_{AP}$ | 0.6 $\mu M$ | Michaelis constant for $PTEN-$induced $Akt$ dephosphorylation; referring to the Michaelis constant for $PTEN-$ mediated $PIP3$ dephosphorylation, $0.1 \sim 0.5 \mu M$ |
| $K_{A_S}$ | 1.47 % | Michaelis constant for $Akt$ activation triggered by growth factors |
| $K_0$ | 0.35 $\mu M$ | Threshold of the total enzyme amount for $Akt$ activation |
| $K_{Akt}$ | 0.2 $\mu M$ | Michaelis constant for $Akt$ dephosphorylation |
| $d_{Mdm2}$ | 0.5 $/h$ | Half-life of MDM2 is about 90 min |
| $d_{MP}$ | 0.1 $/h$ | Degradation rate of $MDM2_p$ is about $5-$fold slower than that of $MDM2$ |
| $d_{MA}$ | 0.6 $/h$ | Degradation rate of the $MDM2-ARF$ complex |
| $d_{MpA}$ | 0.6 $/h$ | Assumed to be the same as $d_{MA}$ |
| $d_{p53}$ | 3.6 $/h$ | Half-life of $p53$ is $5-20$ min |
| $d_{PTEN}$ | 0.5 $/h$ | Half-life of PTEN is longer than 8 h, but $PTEN$ normally undergoes posttranslational modification, which keeps it in an inactive form. |
| $D_{MD}$ | 6 $/h$ | Rate constant for $MDM2-ARF$ disassociation |
| $D_{MpA}$ | 24 $/h$ | Rate constant for $MDM2_p-ARF$ disassociation |
| $n_1$ | 4 | Hill coefficient of $p53-$ dependent expression of $MDM2$ |
| $n_2$ | 3 | Hill coefficient of $p53-$dependent synthesis of $PTEN$ |

## B.5    Comparison of cd-PINN and PINN under Fixed Initial Values or Parameters

In the main text, we highlights the outstanding performance of cd-PINN on generalization. When confined to tasks with fixed initial values or parameters, does the inclusion of continuous dependence into the loss help to improve either the prediction accuracy or the convergence rate of PINN?

For this purpose, we conduct experiments on the LV system – Scenery 1 discussed in the main text. Here we train the PINN model with fixed initial values $R_0 = 8.0$ and $A_0 = 1.0$. The training data set includes 20 real data points and $2^{14}$ residual data points. The same setup is adopted for the cd-PINN. During the training procedure, we first use the Adam optimizer to train for 10000 epochs and then call the LBFGS optimizer for further optimization.

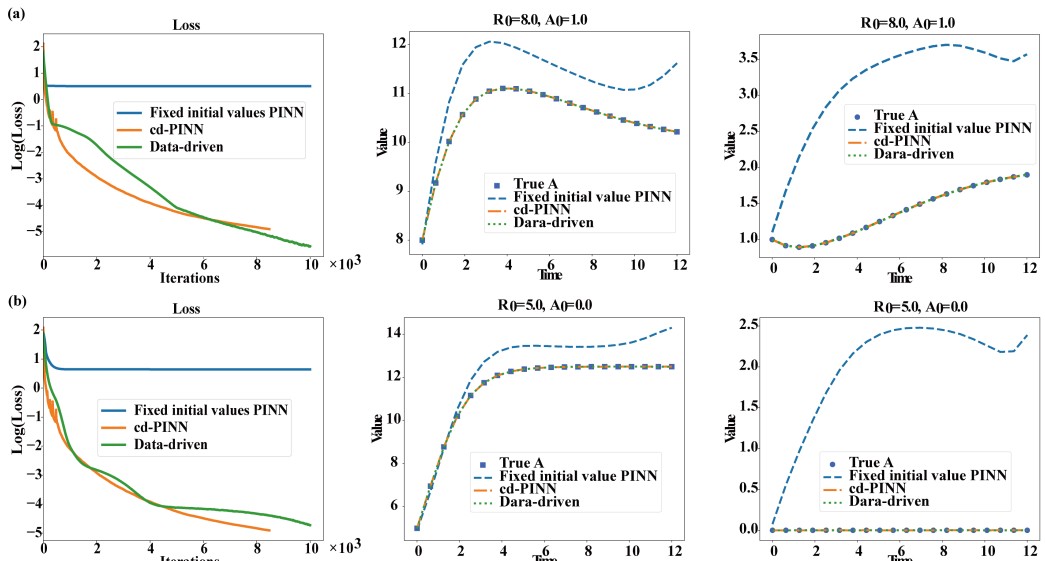

Figure B6: Comparison on the convergence rate and accuracy of cd-PINN, PINN with fixed initial value and a data-regression MLP model. ($a$) and ($b$) illustrate the results for $R_0 = 8.0, A_0 = 1.0$ and $R_0 = 5.0, A_0 = 0.0$ respectively.

The results presented in Figure $B6$ clearly demonstrate that the cd-PINN has a better convergence rate and accuracy than those of PINN. Most astonishingly, in the current case, the convergence rate of cd-PINN is even comparable to a pure data-regression model by MLP. And we believe this is a general feature of cd-PINN, which from the other side highlights the significance of inclusion of continuous dependence.

### B.6   Comparison with Neural ODEs

Neural ODEs(Chen et al. (2018)) are another prominent approach for learning solutions to ordinary differential equations. As a comparison, here we implement the Neural ODE model for the Lotka-Volterra system under Scenery 1 with respect to the same data set and evaluation metrics. The Neural ODE model is numerically integrated by using the 4-order Runge-Kutta method. The model architecture takes the current state $(X(t), Y(t))$ and initial conditions $(X_0, Y_0)$ as inputs, and predicts the state derivatives $(dX/dt, dY/dt)$ as outputs.

From Fig.B7, we can see that the Neural ODE model could effectively capture the system dynamics at the training data points, but shows a much poorer generalization ability than cd-PINN under the testing scenarios ($MSE = 1.37 \times 10^{-2}$ v.s. $4.48 \times 10^{-5}$ for cd-PINN). In addition, the Neural ODE model takes a much longer training time ($25,535s$ for $50,000$ epochs) due to the explicit implementation of RK4 integration steps, whereas PINN and cd-PINN are more computationally efficient.

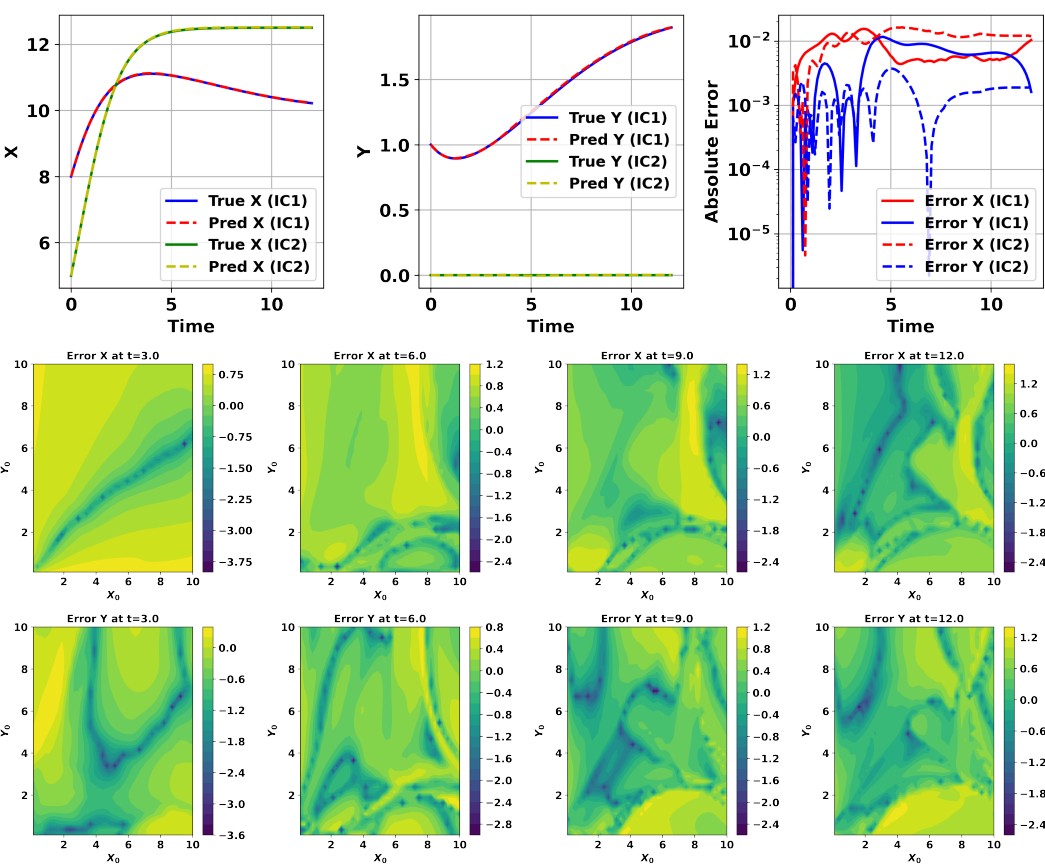

Figure B7: Results of Neural ODEs for the LV model on the training (upper) and test (bottom) data sets separately.

## Appendix C  Implementation Details

We implement cd-PINN using Python 3.6.13 and PyTorch 1.10.1 with CUDA 11.1. Our evaluations are conducted on a machine equipped with Intel(R) Xeon(R) Gold 6132 CPUs and NVIDIA Tesla V100. The source code for this project code is available.

Table C5: Setup of cd-PINN for models tested in this study.

| Problem | Depth | Width | Optimizer | Learning rate | # Iterations | Collocation Points |
|---|---|---|---|---|---|---|
| Logistic equation | 6 | 128 | Adam + L-BFGS | $1 \times 10^{-4}$ | 50,000 | 32,768 |
| LV scenario 1 | 6 | 128 | Adam + L-BFGS | $1 \times 10^{-3}$ | 500 | 32,768 |
| LV scenario 2 | 6 | 128 | Adam + L-BFGS | $1 \times 10^{-4}$ | 100,000 | 32,768 |
| LV scenario 3 | 6 | 128 | Adam + L-BFGS | $1 \times 10^{-3}$ | 500 | 32,768 |
| Underdamped oscillator | 6 | 128 | Adam | $1 \times 10^{-3}$ | 50,000 | 32,768 |
| Critically damped oscillator | 6 | 128 | Adam | $1 \times 10^{-3}$ | 20,000 | 32,768 |
| Overdamped oscillator | 6 | 128 | Adam | $1 \times 10^{-3}$ | 50,000 | 32,768 |
| P53 activation | 6 | 128 | Adam + L-BFGS | $1 \times 10^{-4}$ | 10,000 | 16,384 |