# OpenReview forum: "Incorporating continuous dependence implies better generalization ability"
_ICLR.cc/2025/Conference — Submitted to ICLR 2025_

### Official Review · Reviewer_RhAr · 2024-10-15

**Soundness:** 3
**Presentation:** 3
**Contribution:** 2
**Rating:** 5
**Confidence:** 4

**Summary:**

The authors presented a novel way of introducing continuous dependence into the pinns framework. Specifically for ODEs, they introduced a nn framework with loss designed tailored to the continuous dependence on the initial condition and the hyperparameters. They validated the proposed approach on a range of numerical examples.

While the proposed method is interesting, the referee finds it hard to generalize to more challenging cases, and the literature review is not sufficient. Please see the detailed comments below.

1. The framework is designed and tested for ODEs, but with PDEs, can you reach comparable performance and speed-accuracy tradeoff?
2. The parametric ODEs considered are not challenging enough. Say if the parameters induced singularity or multiscale of ODEs, can the proposed method still work?
3. There has been work in the literature incorporating derivative (smoothness) loss into the loss function and finding that the nns identify smoother functions. I was wondering how the proposed continuity loss compares?
4. Please use more challenging examples.

**Strengths:**

Please see the section above.

**Weaknesses:**

Please see the section above.

**Questions:**

Please see the section above.

---

### Official Review · Reviewer_XYcj · 2024-11-01

**Soundness:** 3
**Presentation:** 3
**Contribution:** 2
**Rating:** 5
**Confidence:** 5

**Summary:**

The paper presents a novel extension of Physics-Informed Neural Networks (PINN), termed continuous dependence PINN (cd-PINN), which aims to enhance the generalization ability of deep learning models applied to differential equations. The authors draw inspiration from the mathematical principle of continuous dependence of solutions on initial values and parameters.

While the idea is straightforward, it introduces additional regularity in parameter dependence. For example, although the parameter \(\mu\) may be continuous, the loss function could impose differentiability conditions on \(\mu\). This is particularly relevant in cases like hyperbolic conservation laws, where the viscosity coefficient can lead to complexities, such as shock locations, that challenge these assumptions. Please discuss the implications of these assumptions, particularly for systems with discontinuities or shocks. This would help clarify the scope and limitations of the method.

The authors demonstrate that cd-PINN achieves significantly higher accuracy than standard PINN through examples like the Logistic model, the Lotka-Volterra model, and damped harmonic oscillators. However, it is worth noting that these examples are relatively simple and may not fully represent the capabilities or limitations of the proposed method in more complex scenarios.

**Strengths:**

The introduction of cd-PINN as a method that incorporates the mathematical framework of continuous dependence

The paper provides compelling numerical results showing that cd-PINN outperforms standard PINN

**Weaknesses:**

The examples selected to demonstrate cd-PINN’s performance are limited to relatively simple cases. Additionally, while the parameter dependence may be continuous, it does not necessarily have to be differentiable. This distinction is crucial, as the lack of differentiability can present challenges in certain contexts, such as in hyperbolic conservation laws, e.g. Burger's equation, where discontinuities like shock locations arise. Please discuss how cd-PINN might handle systems with non-differentiable parameter dependence.

**Questions:**

Including a logarithmic scale in Figure 4 would be beneficial,
Please give a better explanations to your figures on how you compare with other methods

---

### Official Review · Reviewer_tHxD · 2024-11-04

**Soundness:** 3
**Presentation:** 3
**Contribution:** 2
**Rating:** 5
**Confidence:** 4

**Summary:**

In this work, the author proposes a continuous constraint for PINNs, called cd-PINN. This approach allows the ODE problem to continuously depend on its initial conditions and input parameters, thereby transforming it into a family of problems suitable for operator learning. The auther shows theorem on local existence and uniqueness. Experiments on various ODEs demonstrate that cd-PINN performs better on previously unseen conditions.

**Strengths:**

- The author proposes an innovative formulation that converts the ODE to continuously depend on its input parameters.
- This approach combines operator learning with PINN.
- The author provides theoretical results demonstrating the well-defined nature of the simulation.
- Three ODE systems are considered in the study, showing that performance on untrained conditions is typically 1-3 orders of magnitude higher than with standard PINN.

**Weaknesses:**

### Methods
- This formulation seems to only work for inputs in \(\mathbb{R}\) or low-dimensional cases, which limits its applicability to PDE problems, where the input is high dimensional.
- The assumption of continuity may not hold in chaotic systems, such as the Lorenz system.
- Overall, the results are not very significant.

### Writing
- The input parameter \(\mu\) could benefit from further discussion. It would be helpful to include an example alongside Equation (1).
- It would improve clarity to separate the previous formulation of PINN from the newly proposed cd-PINN. Equations (5) and (6) should be moved to a dedicated subsection for PINN.
- For PDEs, it would be beneficial to cite PINO, which combines neural operators with PINNs using pre-training and fine-tuning:
  - Li, Zongyi, et al. "Physics-informed neural operator for learning partial differential equations." *ACM/JMS Journal of Data Science*

**Questions:**

- Could the author give better motivation of application of cd-PINN beyond PINN?
- Can cd-PINN improve the accuracy or convergence rate compared to PINN give a fixed parameters?
- Is it possible to apply the cd-PINN and viscosity in Burgers equation or Reynolds number in Navier-Stokes?

---

### Official Review · Reviewer_itQX · 2024-11-10

**Soundness:** 3
**Presentation:** 2
**Contribution:** 2
**Rating:** 5
**Confidence:** 4

**Summary:**

This paper presents a novel approach, called cd-PINN (continuous dependence physics-informed neural network), aimed at enhancing the generalization ability of deep-learning-based solvers for ordinary differential equations (ODEs). Building on the principle of continuous dependence of ODE solutions on initial values and parameters, cd-PINN extends traditional physics-informed neural networks (PINNs) by incorporating this property to improve performance. By combining neural operators and Meta-PINN techniques, cd-PINN achieves accurate solutions for ODEs with new initial values and parameters, requiring minimal labeled data and no fine-tuning. Experimental results on models like the Logistic model, Lotka-Volterra system, and damped harmonic oscillators show cd-PINN's accuracy is 1-3 orders of magnitude higher than traditional PINNs, with comparable GPU training time. This approach promises enhanced efficiency and accuracy for deep-learning solvers in differential equation applications.

**Strengths:**

**1. Improved Generalization**: By integrating continuous dependence on initial values and parameters, cd-PINN enhances the ability to generalize across different scenarios without additional fine-tuning, which is crucial for solving ODEs in varied applications.

**2. Efficient Data Use**: cd-PINN requires minimal labeled data, reducing the reliance on large datasets while still achieving high accuracy, making it suitable for scenarios where data collection is costly or limited.

**3. Comparable Training Efficiency**: Despite these improvements in accuracy and generalization, cd-PINN maintains a similar GPU time cost as standard PINNs, which suggests it could be implemented without extra computational overhead.

**Weaknesses:**

Although the paper propose interesting strategy, it lacks several points:

**1. Lack of Model Details and Hyperparameter Settings**

The paper does not provide sufficient detail on the architecture and hyperparameter configurations of cd-PINN, such as the number of layers, model parameters, or training configurations. Given that cd-PINN is proposed as a robust solution for a wide variety of differential equations, these details are essential for reproducibility and for evaluating the computational cost of deeper or larger models. Including these specifications would enable researchers to replicate results and better understand how model depth and complexity impact generalization and accuracy.

**2. Limited Scope in Differential Equation Types: Absence of Fundamental PDEs and Comparison with Neural ODEs**

The current focus on ordinary differential equations (ODEs) in the paper feels restrictive, especially given the model’s “PINN” designation, which implies potential applicability to a broader class of partial differential equations (PDEs) involving other derivatives, such as convection-diffusion-reaction (CDR) equations. Extending the study to include such fundamental PDEs would demonstrate cd-PINN’s flexibility and relevance in more complex, real-world applications. Furthermore, comparing cd-PINN’s performance with Neural ODE models would clarify its advantages and limitations, as Neural ODEs are another prominent approach for learning solutions to ordinary differential equations. Including these comparisons would provide a more comprehensive picture of where cd-PINN stands in relation to existing methods.

**3. Extra-interpolation Capabilities**

While cd-PINN claims to learn continuous flow of parameters of ODE, it is unclear whether the model performs well on unseen initial values and parameters. If cd-PINN can indeed generalize to genuinely novel scenarios (e.g., significantly out-of-distribution conditions), this would represent a major strength. Several ablation studies about these cases would be highlight cd-PINN's performance and applicability. Furthermore, a discussion on whether cd-PINN can reliably extrapolate beyond trained distributions would add clarity regarding its practical usability in diverse differential equation settings.

**Questions:**

Please refer to weaknesses section.

---

### Meta-Review · Area_Chair_Mm3f · 2024-12-20

**Metareview:**

The work proposes a method for combining PINNs with operator learning by allowing the network to depend continuously on a parameter of the system. Numerical experiments are carried out on a few simple ODE examples.

**Additional Comments On Reviewer Discussion:**

The scope of the method seems quite limited with parametric dependence being only finite-dimensional (as opposed to functional) and applications examples being only for simple ODE models. Claims that all parametric dependences are continuous are, of course, false and the focus should be on systems where this does hold (as an assumption) as well as to PDE extensions.

---

### Decision · Program_Chairs · 2025-01-22

Reject